# Consumer Behavior Concerning Meat Consumption: Evidence from Brazil

**DOI:** 10.3390/foods12010188

**Published:** 2023-01-01

**Authors:** Claudimar Pereira da Veiga, Mirian Natali Blézins Moreira, Cássia Rita Pereira da Veiga, Alceu Souza, Zhaohui Su

**Affiliations:** 1Fundação Dom Cabral—FDC, 760 Princesa Diana, Alphaville, Lagoa dos Ingleses, Nova Lima 34018-006, MG, Brazil; 2School of Management, Federal University of Paraná, Av. Lothtário Meissner, 632, Curitiba 80210-170, PR, Brazil; 3Department of Health Management, School of Nursing, Federal University of Minas Gerais, 190 Alfredo Balena, Belo Horizonte 30130-100, MG, Brazil; 4Business School—PPAD, Pontifical Catholic University of Parana, Imaculada Conceição 1155, Curitiba 80215-901, PR, Brazil; 5School of Public Health, Institute for Human rights, Southeast University, Nanjing 210009, China

**Keywords:** reduction of meat consumption, consumer behavior, environment, health, animal welfare

## Abstract

Recent research has consistently related the production and consumption of meat with environmental degradation, health problems, and damage to animal welfare. However, meat consumption represents a well-established eating behavior among many consumers. After all, meat is a central food in many cultures, and changing eating habits can be very challenging. Nevertheless, there is a group of consumers who have reduced the consumption of meat in their diet. Understanding the facilitators and barriers that influence these behavioral changes in different cultures and contexts can help to promote future collective reductions in meat consumption. This article investigates the main facilitators of and barriers to the reduction of consumption in the largest meat-consuming market in the world to contribute to the advancement of knowledge on meat-reduced diets. In order to achieve this objective, semi-structured interviews were conducted with consumers who reduced their meat consumption. In this context, a content analysis was conducted to identify 22 facilitators and 15 barriers classified according to the COM-B system. This system conceptualizes Behavior, which can be influenced by Capability, Opportunity, and Motivation. The results of this research corroborate previous discoveries or confirm the presence of a set of facilitators of and barriers to the reduction of meat consumption previously suggested in the literature.

## 1. Introduction

Consumer behavior concerning the consumption of food products has attracted the attention of the academic community and companies [1], especially regarding cultural [2], economic [3], environmental [4], social [5], and human health aspects [6,7]. Given the impact of meat production and consumption on environmental sustainability, animal suffering, and human health, the recent attention of researchers on the topic is not surprising.

Regarding the eating behavior of meat consumption, recent research has pointed out some problems associated with environmental degradation [8], health problems, and compromised animal welfare [9,10,11]. In light of these issues, this has been one of the central subjects of recent research in food science. The literature shows a growing concern for understanding the current status of research and future direction on the adoption of food alternatives that generate a reduction in meat consumption to face important health and sustainability challenges [10,12]. On the other hand, considering the central role of meat in the food routine and the various hedonic, social and cultural aspects linked to this food, many consumers may not be willing to eliminate this component from their meals [13].

Promoting the global reduction of meat consumption will undoubtedly be challenging, as it is necessary and urgent [6]. On the other hand, although there are many challenges, meat consumption is a voluntary behavior that can be modified [14]. For example, some consumers, known as flexitarians, have transitioned to more sustainable eating styles, reducing the frequency of meat consumption rather than completely removing it from their diets [15,16]. Although the complete elimination of meat seems a distant reality, intermediate behaviors such as flexitarianism have gained popularity in the food routine [17].

Research related to the frequency of meat consumption and determinants of meat intake [6,10,12] is also highlighted by the previous literature. For example, research in Slovenia showed that educational level and status affect meat consumption frequency and sustainable attitudes [6]. The authors suggest that public health and environmental campaigns that reduce the frequency of meat consumption should focus on the group with the least susceptibility to such a change, that is, on individuals with a low educational level [6]. The level of education was also a determining factor in reducing meat consumption in Germany, as well as demographic factors such as gender and age [12]. Likewise, there is evidence that consumers from western countries with low socioeconomic status tend to eat more meat and buy lower-priced meat products [10]. That research has demonstrated that consumers’ attitudes and capacities for dietary change are strongly influenced by their socioeconomic position, suggesting the need for further studies on the subject [10].

Despite the importance of the theme, research on changing the behavior of consumers who reduce meat consumption or adhere to flexitarianism [18] is limited and restricted to a few countries, such as the Netherlands [16], Australia [19], New Zealand [20], France and Germany [21], and Poland and Slovakia [22]. In Brazil, considered the third country in the world ranking for meat consumption, the consumption per capita is significant (24,443 kg), second only to Argentina (36,035 kg) and the United States (25,969 kg) [23]. Therefore, further studies are needed to understand consumer behavior concerning reducing meat consumption. [6,14,15,16,17]. In this respect, the present study aims to apply and analyze, from the COM-B system, presented in Figure 1, a framework for understanding behavior [24], the main facilitators and barriers faced by consumers who have reduced meat consumption in Brazil.

The change in consumer behavior identified in the literature shows that the cultural domain of meat consumption may be less evident [16]. Review studies have shown that such behavior changes can be understood by evaluating the interactions between capabilities, opportunities, and motivations [24,25]. Research to better understand perceived barriers and motivations related to meat consumption should consider cultural and economic aspects [3].

## 2. Materials and Methods

This is a qualitative and in-depth study using semi-structured interviews. In this respect, it is also relevant to show that the current research is part of a (recent) line of qualitative studies in meat reduction and meat substitutes (see, e.g., [26,27,28,29,30]). In this context, the COM-B Model [24] was used to analyze consumer behavior. The applied model presents the behavior generated by interactions between (i) capabilities, (ii) opportunities, and (iii) motivations (see Figure 1).

According to the model, (i) capabilities can be physical or psychological and are related to the knowledge and skills needed to engage in a specific behavior. An (ii) opportunity, in turn, is represented by social or environmental factors that are outside the individual and drive behavior. Finally, (iii) motivation represents the mental processes (reflective or automatic) that drive behaviors [24].

The COM-B Model was selected to identify the facilitators and barriers that could be presented according to the dimensions in light of this theory. This model has been used and validated by several studies in the literature, such as that of Graça et al. [25]. The literature highlights that this model aids in the identification of facilitators and barriers to reducing meat consumption. By suggesting facilitators and barriers grouped in the three dimensions of the model, the objective is to provide a mapping of facilitators and barriers of (i) capability, (ii) opportunity, and (iii) motivation for the adoption of diets with less meat in Brazil.

### 2.1. Questionnaire Development

This study was performed using a questionnaire and semi-structured interviews. The semi-structured interviews were conducted according to a script based on previous literature and the main barriers and facilitators for reducing meat consumption, presented in Figure 2. The questionnaire was composed of two parts. The first part identified the sociodemographic aspects with four questions. The second part consisted of 16 questions related to consumer behavior. The first version of the questionnaire was submitted for a pre-test to assess the content validation of the research instruments before the final application. In addition, the pre-test audios were recorded, transcribed, and analyzed, which allowed minor adjustments to be made in the final questionnaire. The questionnaire used as a script in the interviews can be found in Appendix A.

### 2.2. Data Collection

The study sample consisted of 18 people who were reducing their meat consumption. This target audience is still small in Brazil. Although previous studies have investigated the intention to reduce meat consumption [22], this does not always lead to a fundamental change in behavior. Thus, the inclusion criteria for consumers participating in the research were to be 18 or older, living in Brazil, and reducing meat consumption at the time of the interview. Like Neff et al. [31], the research focused on the participants’ interpretations of meat consumption. Therefore, the participants were not required to have a pre-defined target of reduced meat consumption, leaving the consumers to self-classify. Likewise, the participants were selected regardless of the specific type of meat they consumed (e.g., pork, beef, lamb, chicken, fish, and seafood). The main exclusion criterion for participation in the study was vegetarian or vegan eating behavior since the intention was to identify facilitators and barriers for reducing meat consumption rather than wholly abstaining from it.

Snowball sampling was employed in this work, using the researchers’ contact network, disseminating on social media, and sending invitations to universities in Brazil. Regarding ethical issues, all the participants received information about the research through an informed consent form and voluntarily agreed to participate. The interviews took place virtually through web conferencing tools and were recorded, transcribed, and exported to ATLAS. ti web content analysis software.

### 2.3. Content Analysis

The collected content was subjected to content analysis in six steps to identify patterns [32], as detailed in Figure 3. The (i) first phase involved prior reading for recognition and familiarization with the data, thus allowing an (ii) inductive coding in the second phase. In phase three, the (iii) generated codes were grouped based on similarity and previous literature. In phase four, (iv) facilitators or barriers mentioned by fewer than three participants were disregarded. In this phase, a new deductive coding was also used to classify the facilitators and barriers identified in broader themes according to the COM-B Model, Capabilities, Opportunities and Motivations. In phase five, the (v) definitive naming of the themes and the removal of extracts from the interviews that exemplified each identified theme was carried out. The last phase involved the (vi) drafting of the report. In order to enable a better understanding, the results were stratified based on patterns related to potential facilitators and barriers of capability, opportunity, and motivation to reduce meat consumption in Brazil.

## 3. Results

Eighteen in-depth interviews were conducted between 4 July and 4 September 2021. As suggested by the ethics council, each participant received a fictitious name to preserve their identity. The duration of the interviews varied, with the average interview lasting between 21 and 46 min. In the descriptive statistics analysis, it was observed that 14 participants identified themselves as women, and 4 identified themselves as men. Their ages ranged from 19 to 53 years. As for their places of residence in Brazil, 9 participants lived in the South region, 3 in the Southeast region, 3 in the North region, 2 in the Northeast region, and 1 in the Midwest region. Regarding education, 12 participants had completed higher education, and 6 had incomplete higher education. At the time of the interviews, the participants’ meat consumption varied from occasional consumption (only at parties or events, for example) to daily consumption (with reduced portions of meat compared to previous consumption levels). The conscious reduction in meat consumption duration ranged from 3 months to 12 years. An overview of all the facilitators and barriers identified in the study can be found in Figure 2.

Twenty-two facilitators for reducing meat consumption were identified, of which eight had not been described in the previous literature, which is an essential contribution of this study: (i) health problems or dietary restrictions (C.F.); (ii) acquisition of new knowledge, skills, and information (C.R.); (iii) cooking or buying their own food (O.F.); (iv) access to restaurants and fairs (O.F.); (v) access to a nutritionist (O.F.); (vi) feeling of flexibility in the diet (M.R.); (vii) dislike of cooking meat (M.A.), and (viii) habit of eating little meat (M.A.). Additionally, four barriers identified in this study were also not previously described in the literature: (i) health problems or dietary restrictions (C.F.); (ii) price of substitutes (O.F.); (iii) meat availability (O.F.); and (iv) social interaction (O.S.)

### 3.1. Facilitators

The facilitators and barriers for reducing meat consumption in Brazil identified in the study were classified as (i) physical capability, (ii) psychological capability, (iii) physical opportunity, (iv) social opportunity, (v) reflexive motivation, and (vi) automatic motivation.

#### 3.1.1. Physical Capability

Content analysis showed that some health issues or dietary restrictions potentially drove respondents to reduce their meat consumption. However, it is important to emphasize that the health problem or dietary restriction was not always directly related to meat consumption. Nevertheless, they motivated the participants to review their eating behavior, helping to reduce meat consumption. Participant Mateus, 30, for example, told how his lactose intolerance led him to become interested in vegan meals: “*I also became interested in this diet that vegans follow and what they do because there is no lactose in it, and so for me that was a sure thing, I was sure that I wouldn’t be consuming lactose*”.

Other participants reported problems that directly impacted their meat consumption. This is the case of Leonardo, 21 years old, for example, who is “*allergic to some kinds of meat, like pork and shrimp*”.

#### 3.1.2. Psychological Capability

The acquisition of new knowledge and skills, as well as information, mainly nutritional and culinary, seems to be critical for consumers to be able to make a transition to diets with lower meat consumption. Consumers usually acquire these skills through information published on the internet, through information provided by nutritionists, or in courses. In particular, the ability to learn new meatless recipes appears to spur changes in consumer eating behavior. Amanda, 34, for example, explained this as follows:

*“I think what worked for me and helped me, and might help other people, is getting familiar with recipes. Who knows, on social media, social media profiles, YouTube channels that promote vegan and vegetarian recipes. Because there you can find recipes and you can make cool food that doesn’t have any meat in it, and I think that is what helped me most”*.

Obtaining nutritional knowledge, in turn, also plays an important role in reducing meat consumption, as it helps the consumer to feel safe when making food exchanges. Ricardo, 27, for example, said:

*“I went to a nutritionist, questioned him and asked things […] he then helped me a lot to understand how to put a menu together, replacing a certain spice and why it’s important to do that, and other kinds of food that have proteins similar to meat”*.

#### 3.1.3. Physical Opportunity

Content analysis has shown that the high cost of meat can also drive a change in eating behavior. Participant Jessica, 23 years old, for example, reported that

*“meat these days is really, really expensive! So, whether you like it or not, a diet based on plants, on grains, ends up being a little bit… cheaper”*.

The act of cooking or buying one’s own food also proved to be an important facilitator for reductions in meat consumption. The participants seem to have greater freedom of choice to make changes in their eating behavior when they are responsible for preparing or purchasing their food. Patricia, 52, for example, commented that “*now that I’m in charge of the stove, the fridge and all that, I’m finding it easier [to reduce my meat intake], right?*”.

Access to restaurants and fruit and vegetable markets was also frequently cited by the respondents as a facilitator to reduce meat consumption. In the case of restaurants, it seems important for consumers to have access to places that offer meat-free meals that are tasty, nutritious and affordable. Carolina, 19, for example, commented:

*“One really cool thing that I saw at my university is that, for instance, the canteen, the university restaurant there, has a vegetarian option that varies a lot every day of the week and is regulated by a nutritionist… So, having that option available at universities, in schools, makes it easier”*.

Fruit and vegetable markets, in turn, are a place to buy a wider range of vegetables, helping to replace meat with other food components. Matthew, 30, for example, stated:

*“You know, I think the fruit and veg market helped me a lot. Doing the shopping there. I mean, I said to myself ‘today, I want to get this at the market… I want to get something that I don’t usually eat’, and then I’d get it”*.

Access to nutritionists is another potential facilitator for those who wish to reduce or who are already reducing their meat consumption. These professionals can help consumers to choose appropriate food substitutes and feel safe about their health. Helen, 37, for example, said her nutritionist *“gave her a vegan protein”* to help her *meet “the need for protein”*.

#### 3.1.4. Social Opportunity

The respondents stated that the influence or support of people close to them represents an important facilitator for reducing meat consumption. Having contact with friends or family who are already vegans or vegetarians or being able to count on the support of others in their eating behavior change seems to create a favorable and welcoming social environment for meat-reducing consumers. Leonardo, 21, for example, said:

*“I think that influence has something to do with it… I have friends who also do this with their food or meals, I mean, with less meat, and I think that this ended up making me a bit more of a vegetarian and vegan, so that helps as well”*.

The endorsement of opinion leaders also represents a potential enabler for reduced meat consumption, with athletes, online influencers, and celebrities inspiring consumers to continue the food transition process. Eliza, 29, for example, stated that,

*“I go rock climbing and the person that I think was the best climber at the time is a vegetarian. So, I started to look for high performance people who are vegetarians”*.

#### 3.1.5. Reflexive Motivation

Some interviewees commented on the perception of better health performance after adhering to diets with less meat consumption. They reported improved digestion, greater physical disposition, better sleep quality, and weight loss, among other perceived benefits. Eliza, 29, for example, commented:

*“I feel like doing more exercise, I feel better eating less meat, and I feel healthier in general. So, it’s good for… for everything, for your self-esteem, for… I feel less tired and I digest food more quickly”*.

Some participants also spoke about their animal welfare concerns regarding meat consumption. Roberta, 29, for example, talked about “*the way that animals are treated as well, right? It’s really cruel at the slaughterhouse, you know? It’s a life that goes at the slaughterhouse*”. However, it is worth noting that although consumers show concern for animals, this motivation alone may not be strong enough to generate a lasting behavior change. Isabela, 26, for example, stated that although she went some time without eating meat after watching a documentary, today, she is “*aware of how animals are treated*”, but “*even so*”, she eats meat.

In addition to the concern for animal welfare, the participants also claimed that the change in eating behavior is related to concern for their own health. A comment made by Gustavo, 43, exemplified this issue:

*“I began to rethink my diet a lot, not only regarding meat but also things that I eat that don’t have any meat in them. Mainly cutting out, reducing, eliminating ready-made processed food, things that have a lot of processed spices and a lot of preservatives in them, a lot of artificial flavoring. So, I cut down on these things a lot and as a result I also cut down on my meat consumption”*.

In line with the previous literature, concern for the environment as a motivator for reducing meat consumption was identified among the interviewees. Roberta, 29, for example, spoke: “*of cattle involving… widespread deforestation to make way for pasture and also of the gases produced by cattle that affect the environment*”. Notably, reflexive motivation stemming from an environmental concern does not often seem to be the main reason why consumers reduce meat. Concern about meat-related environmental issues may even only arise after the reduction begins. Elaine, 22, for example, said she initially cut down on meat because she did not like the texture and only later began to “*develop a greater interest in sustainability issues*”. Thus, despite environmental awareness being a potential facilitator for reducing meat consumption, in some cases, concern about environmental issues may only arise after the change in behavior has begun and depends on the information made available to the consumer.

Informed consumers may have a greater incentive to change their eating behavior. The sources of this information can be diverse, such as formal courses, documentaries, social media, and health professionals. Participant Suelen, 21, for example, exemplified the role of studies and documentaries in providing information by stating that

*“I learned quite a few things through courses, documentaries, and classes at university that made me see that I wanted to reduce [meat] consumption in my diet”*.

On the other hand, even for an informed consumer, completely abandoning a meat-based diet is an arduous task. The results showed that consumers who choose to reduce meat consumption experience a sense of flexibility. That is, by not identifying as vegetarians or vegans, they are free to consume meat if and when they wish. Gustavo, age 43, for example, stated: “*I’ve reached the point where I can still satisfy my taste buds and hunger for food that tastes good, but also look after my health with the effects, you know, the results, that I have seen, right?*”. Some participants also believed that a reduction was easier than the total elimination of meat from their diet. Amanda, 34, commented that “*cutting down wasn’t so hard*”, but she believes that “*it would have been hard to cut out meat completely*”.

Finally, this study also identified factors related to convenience and collective feeling as facilitators for reducing meat consumption. Leonardo, 21, for example, commented: “*I would say that eating less meat seemed a little more practical, especially in this remote format [owing to the COVID-19 pandemic] when I have to cook more and end up eating more because I’m staying at home. So, sometimes it’s more practical to replace meat with some broccoli, a boiled egg, perhaps some salad or grains in general*”. There is a positive relationship between the feeling of doing the right thing and changing eating behavior [33]. Helen, 37, reported that after reflecting on meat consumption, she concluded: “*I began to think that I really had to do it and cut down on meat*”. However, it is worth noting that although consumers relate their reduction in meat consumption to a collective benefit for society, there is a feeling that the behavior change is of low impact. Roberta, 29, for example, reported that reducing meat consumption *“may ease your mind, but not very much, not very much*”.

#### 3.1.6. Automatic Motivation

The content analysis showed that some participants had an automatic motivation to change their eating behavior: a natural lack of interest in meat consumption. Elaine, 22, for example, stated that “*I could go without red meat and chicken, and even fish to tell you the truth*”. The results also demonstrate that not appreciating the attributes of meat-based foods, such as taste, texture, smell, or appearance, can lead consumers to avoid this component in their diet automatically. Elaine, 22, for example, declared, “*I didn’t even like the texture of meat, and so I began to stop eating it*”.

Participants who do not like to cook meat also seem to have automatic motivations to change their eating behavior. Some of the interviewees find that the act of handling meat to prepare it can be unpleasant or disgusting. Helen, 37, for example, stated that “*If I have to fry meat, spice it, put my hands on meat… that’s awful, I don’t want to do it!*”.

Additionally, not having the habit of eating meat seems to make eating behavior change more viable for consumers. Aline, 33, for example, said she has never “*been one who ate a lot of meat*”, whereas Marisa, 53, also said that she had never been “*much of a meat eater*”. Respondents also showed interest in trying new types of food. This interest may arise before the reduction in meat consumption or after the change in eating behavior. Patricia, 52, for example, said:


*“I think I started experimenting more, you know? So, every now and then I go to order food on iFood, then I like to order different things that I think I might like, you know? So, instead of ordering the usual, I take more risks, because I’m trying to identify new kinds of food, right?”*


Other participants demonstrated that changing eating behavior is facilitated by preparing beautiful, attractive, and tasty food that does not use meat-based elements. For example, Gustavo, age 43, commented that, through living with a vegetarian, he learned that *“we have a lot of tasty things in vegetarian cuisine or in a cuisine that does not necessarily use meat”*.

### 3.2. Barriers

#### 3.2.1. Physical Capability

Some health problems or dietary restrictions represent an important barrier to reducing meat consumption, especially regarding the substitution of this food. Aline, age 33, for example, stated that:

*“I find it hard to digest grains, no matter how I cook them I just can’t”. *Thus,* “no matter how much people want to make this change in their eating habits, some health issue prevents it”*.(Carolina, age 19)

#### 3.2.2. Psychological Capability

As preparing meat substitute foods requires acquiring new knowledge, skills, and information, some respondents expressed concern about learning new culinary skills, especially related to meatless recipes, as well as learning to replace meat in their diet correctly. Roberta, 29, for example, when asked about this issue, commented:

“*I think it’s all about recipes. I still find it hard to think about food without animal proteins*”. Carolina, aged 19, in turn, said:

*“I think it’s about a lack of information on how to keep the same nutrients because a lot of people think ‘ah, protein is only found in meat’. So, there is a lack of information on how you can keep the same level of proteins but with a different diet”*.

#### 3.2.3. Physical Opportunity

Regarding physical opportunities, the respondents commented that the high prices of plant-based meats and other vegan and vegetarian options discourage the purchase of these products. Amanda, 34, for example, commented, “*When it comes to sausages, I think like this, I would like to be able to not eat sausages with meat. I’d like to eat sausages with TSP [textured soy protein], but I don’t buy them because they’re much more expensive*”. In addition, meatless meals or products for preparing meatless meals are available in few commercial establishments and with little variety. Suelen, 21, for example, commented that, “*replacing food is also very hard because you go to the market and they don’t have anything […]. You can’t even find TSP [textured soy protein] at the market, which is pretty basic stuff, right? It’s not easy*”. In this context, the convenience related to the greater availability of meat-based meals represents an important barrier for consumers seeking to change their eating behavior. Marisa, 53, exemplifies how this occurs. “*You end up eating it because it’s right in front of you, right? You’re hungry and you eat it*”.

#### 3.2.4. Social Opportunity

Meat-based food is part of the food routine of many Brazilians. From this perspective, social interaction is a barrier for consumers who live with people who consume meat regularly. Helen, 37, provided an example of the meat-free challenge of living with her son:

*“There are only the two of us at home. Am I going to make food and force him to do without meat when he wants to eat it? I end up eating it with him”*. Another barrier identified in the interviews is related to the culture and tradition of meat consumption. After all, it is a central component of the meals of many Brazilians, and the social norm determines the consumption of meat as a standard. Leonardo, 21, for example, said that *“It’s already kind of established in society that we need to eat meat and eat it in large quantities since childhood”. He thinks that “this could be a latent difficulty”*. Other consumers also commented on the barbecue culture: *“you see it in commercials, you see it in Brazilian culture, people having barbecues and all that”*.(Ricardo, age 27)

Because of the culture and tradition of consuming meat-based meals, some participants also felt that changing their eating behavior could harm them socially. Matthew, 30, offered an example when recounting his experience with social events:

*“Sometimes there was an event at the company when I was still working and I had to go, I had to go because everyone went and there was only meat, there was nothing that a vegetarian could eat […]. And when you didn’t eat anything, you still felt bad, because people sometimes either made a joke or didn’t say anything, but they thought you were… you know? It’s a bit of a bad feeling like that, on both sides. People are not prepared to deal with it and so are you sometimes… you don’t want to go through this, so sometimes you prefer to avoid this situation […]. I don’t go to that kind of event anymore. Especially if it’s a barbecue, for example”*.

The social harm faced by people who seek to reduce their meat consumption is also due to the reluctance of those close to them. In this respect, consumers may have to deal with opposition, especially from family members, to their behavior change. Eliza, 29, for example, made the following statement about her mother:

*“She thinks it’s ridiculous to eat less meat or cut out meat altogether. So, this year, the meat I’ve consumed wasn’t bought by me. I haven’t bought any meat this year. It was all meat that my mother bought”*.

#### 3.2.5. Reflexive Motivation

A barrier related to reflexive motivation cited by the interviewees was the perception of a lack of convenience in obtaining substitute foods or preparing meatless meals. Suelen, 21, commented that “*sometimes, meat is much more practical. You buy a packet of chicken wings and make it. So, it’s kind of difficult, this issue of researching what to eat and sometimes I don’t have time to make food as well*”. Furthermore, Ricardo, age 27, also said that it could be difficult “*to find food and learn how to buy it. This was an important point: learning to buy!*”. It can be seen that a change to a non-standard diet requires additional effort on the part of consumers, who may be discouraged from changing their eating behavior.

Additionally, there is a health concern with regard to consuming a low-meat diet. As meat is the main source of protein in a western diet [34], consumers often believe that its absence can lead to nutritional deficiencies [2]. Jessica, 23, for example, spoke of her experience and the concerns reported by her mother: “*only, my mother always says, ‘your health, you have to look after your health, with vitamins and all that, protein’, you know*”.

#### 3.2.6. Automatic Motivation

Some of the interviewees stated that they enjoyed the taste of meat and that they felt the desire to eat it, demonstrating that they had hedonic feelings towards meat. Isabela, 26, for example, said: “*Meat is really good, isn’t it? So, it’s hard to control the urge to eat it because I really like eating meat. It’s tasty*”. Some consumers have also been shown to have developed an attachment to or dependence on meat consumption. Mateus, age 30, offers an example of this:

*“Baloney was something we ate a lot of when we were kids, because it was cheap. So, I eat this stuff from time to time, but I eat it with a guilty conscience, knowing that I’m eating something that’s not good for me, you know? But for the taste itself, just to satisfy that urge. Maybe I’ll be on my deathbed and I’ll be like, ‘Oh, I want to eat a sandwich’ just because I miss that sandwich, you know?”*.

Likewise, the frequent habit of eating meat represents a barrier in the interviewees’ food transition journey. Leonardo, 21, said that “*since childhood we have been used to eating a lot of meat, so to stop eating it as adults is hard*”. Laura, 26, also believes that a barrier to be overcome is “*this habit of eating meat since we were children*”.

In addition to consuming meat-based diets, the respondents claimed that a lack of interest in trying meals without meat represents a barrier to changing eating behavior. Typically, the respondents did not attribute this barrier to themselves but noted that other people in their environment are resistant to non-meat meals. Taís, 26, for example, stated that “*anyone who didn’t want to stop [eating meat] kind of closed themselves off from things other than meat*”. The participants were convinced that meatless meals can be very tasty and assumed that other people would also notice this if they were willing to try it.

## 4. Discussion

The results revealed that multiple facilitators and barriers of capability, opportunity, and motivation influence the eating behavior of Brazilian consumers about reducing meat consumption. There are similarities and dissimilarities between what has been described in previous studies in the literature. Achieving future reductions in meat consumption will be challenging for all parties involved, both consumers and external agents. However, even though there are challenges, the integrated understanding of facilitators and barriers, as proposed in this study, offers insights that can assist in developing promising strategies for collective meat reductions in Brazil or other regions.

Figure 4 highlights one of the main findings of the research in terms of actions that can be developed by various agents other than consumers, such as policymakers and companies, to promote reductions in meat consumption.

Figure 5, in turn, offers examples of actions that can be taken by consumers who intend to begin or continue reducing their meat consumption. By applying the COM-B Model, Figure 4 and Figure 5 demonstrate actions that various stakeholders can develop to assist consumers in developing capabilities, opportunities, and motivations for reducing meat consumption based on the findings of this study. In addition, examples of actions that consumers themselves can take are also provided. The intention behind giving these examples is not to exhaust the possibilities for action but to exemplify how knowledge of facilitators and barriers of capability, opportunity, and motivation can further the adoption of diets with less meat.

The influence that the food context exerts on consumer choices can be seen. Therefore, providing infrastructure to support plant-based diets may be a promising option for encouraging a reduction in meat consumption [35]. Another point is related to the importance of the role of health professionals in providing consumers with guidance concerning alternative protein sources [19]. Previous studies have suggested that people who successfully put meat reduction into practice often mention a close person or opinion leader who has also made similar changes to their diet [36]. These opinion leaders can help consumers feel they are on the right track, even reducing the effects of social pressures to eat meat [35].

The participants in the survey sample were satisfied with the improvements they noticed in their health when adopting reduced-meat diets. The reports of these consumers who are realizing in practice the benefits of the changes they have made to their diets could be used to promote diets with less meat. In this context, the reducers’ experience in the face of an internal conflict can act as an invitation for other consumers to try reducing meat consumption to obtain the same benefits.

Previous studies have suggested that people may face internal conflicts regarding meat consumption since it is possible to feel love for meat and love for animals at the same time [37]. On the other hand, other studies have already suggested that consumers are increasingly aware of the relationship between health and food [38]. That evidence has shown that promoting health benefits can be one of the strongest arguments for promoting low-meat diets [20]. These authors suggest that future efforts to reduce meat should include disseminating clear and consistent information on the various impacts of food choices. After all, if consumers are not aware of the benefits of change, why are they going to change?

Furthermore, science plays a particular role in information, as consumers may demand scientific evidence before considering a reduction in meat consumption to benefit the environment [39]. Therefore, it is a guideline for researchers and policymakers to develop compelling, evidence-based arguments for promoting low-meat diets. This finding indicates that consumers may better accept a recommendation to reduce meat rather than adopt vegetarian or vegan diets. Consequently, a suggestion for future campaigns is to promote a gradual and flexible reduction in meat consumption, emphasizing to the consumer that he does not necessarily need to eliminate meat all of a sudden but can take small steps toward a healthier, sustainable, and animal-friendly diet.

In Brazil, for example, there is “Segunda Sem Carne” (Meatless Monday), a social campaign that approaches the reduction of meat in a “light” way, inviting consumers to abstain from meat one day a week [40]. More campaigns with similar approaches could be constructive. A previous study showed that even after eating meat, retired people over 60 had less of an appetite for meat [3]. Another study previously carried out by Weinrich [41] with German, French and Dutch consumers also indicated that consumers greatly appreciated the taste of meat in these three countries.

A possible alternative to lower this barrier is the promotion of tasty meatless recipes. The objective of this strategy is to create a counterpoint related to hedonism. People fond of meaty meals will hardly stop liking meat, but that does not prevent them from developing hedonic feelings for meatless meals. This can lead, albeit moderately, to substituting meaty meals with meatless meals that are also appetizing. Another suitable strategy is to expand the offer of alternative protein options in commercial establishments. This could help to familiarize consumers with tasty alternatives prepared by chefs who know how to enhance the flavor of these foods [42].

It has been noticed that an attachment to meat can represent a difficult barrier to overcome due to the connection created between the consumer and meat. A previous study suggested that people who have formed a connection with meat are less willing to reduce meat consumption and follow a more plant-based diet [43,44,45]. However, as difficult as it may be, it is not an insurmountable barrier, as shown by some participants in this study, who, despite being attached to meat, were managing to reduce its consumption.

## 5. Conclusions

This study aimed to identify facilitators and barriers to reducing meat consumption in Brazil. Data collected through questionnaires and semi-structured interviews were subjected to content analysis to achieve this objective. As a result, 22 facilitators and 15 barriers of capability, opportunity, and motivation were identified that potentially influenced the reduction of meat consumption.

In general, it can be said that all the factors discussed in this study work together to change eating behavior and maintain this practice. For example, when the interviewees were asked about the reason for the reduction in their meat consumption, they all responded with a set of factors, demonstrating that lower meat consumption occurs after the consumer accumulates several facilitators and overcomes several barriers in terms of capabilities, opportunities and motivations that make it possible to change eating behavior.

### 5.1. Theoretical Contributions

As a theoretical contribution, this work reinforced previous research findings by confirming the presence of facilitators and barriers to reducing meat consumption previously suggested in the literature. This research also proposed and discussed eight new potential facilitators not been described in the previous literature, which is an essential contribution of this study: (i) health problems or dietary restrictions; (ii) acquisition of new knowledge, skills, and information; (iii) cooking or buying their own food; (iv) access to restaurants and fairs; (v) access to a nutritionist; (vi) feeling of flexibility in the diet; (vii) dislike of cooking meat, and (viii) habit of eating little meat. Additionally, four barriers identified in this study were also not previously described in the literature: (i) health problems or dietary restrictions; (ii) price of substitutes; (iii) meat availability, and (iv) social interaction. Hopefully, this set of proposed new facilitators and barriers of capability, opportunity, and motivation will guide future research that will become part of the growing body of literature on reducing meat consumption.

### 5.2. Managerial Contributions

From a practical point of view, this work offered information that various stakeholders can use in reducing meat consumption since it is expected to be necessary to rely on multiple capabilities, opportunities, and motivations for the feasibility of substantial reductions in Brazilian people’s meat consumption. This will require joint efforts from various civil, economic, political, and market actors, such as consumers, researchers, NGOs, public entities, industry, retailers, restaurants, and opinion leaders. After all, as possible generalized reductions in meat consumption would generate collective benefits, the responsibility to make efforts to this end lies with everyone.

### 5.3. Limitations and Future Research

This study has some limitations, mainly due to its sample size, meaning that generalizations cannot be made. In addition, the data collected were based on self-reported behaviors, which do not always accurately reflect the actual behavior of consumers. It is also important to note that the study was carried out during the COVID-19 pandemic period, a period characterized by significant changes in consumer habits and behaviors.

Brazil is among the world’s largest meat consumer market countries (23, 30), and a more holistic view of the eating behavior of Brazilian consumers stands out. In this context, future research could investigate consumers who want to reduce meat consumption but have not yet managed to change their eating behavior, as well as Brazilian consumers who do not intend to make such a change. Both approaches could help identify new barriers to reducing meat consumption in Brazil and other countries. Additionally, given Brazil’s continental dimensions and cultural diversity, further studies could investigate the facilitators and barriers to reducing meat consumption in different geographical regions. A more segmented investigation would allow the planning and application of more targeted interventions. It also highlights the need for future studies that address the importance of reducing meat consumption for humans, animals, and the planet.

Finally, this study did not allow for an in-depth analysis of each of the dimensions of the COM-B model. Therefore, further studies could focus on a single dimension for a more detailed assessment. This work was not intended to exhaust a topic of global importance but to pave the way for future theoretical, managerial, and political discussions.

## Figures and Tables

**Figure 1 foods-12-00188-f001:**
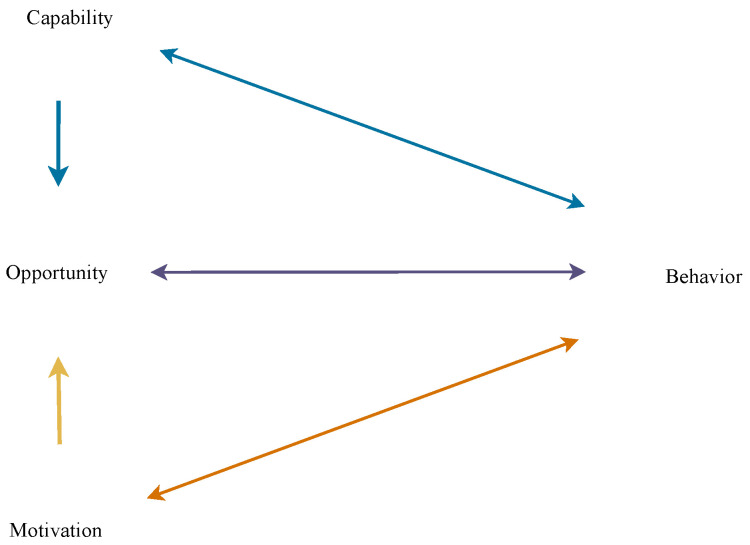
The COM-B system. Source: adapted from Michie et al. [24] (p. 4). Note: The arrows represent the potential influence between the proposed elements.

**Figure 2 foods-12-00188-f002:**
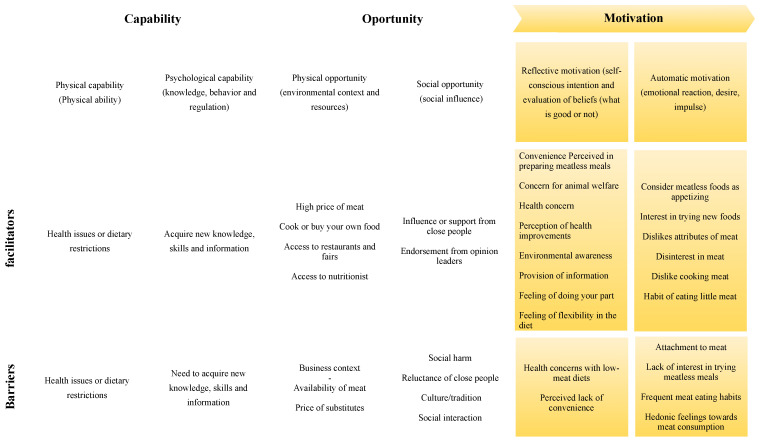
Facilitators and barriers for reducing meat consumption in Brazil. Note: Classification based on the study by Graça et al. [25].

**Figure 3 foods-12-00188-f003:**
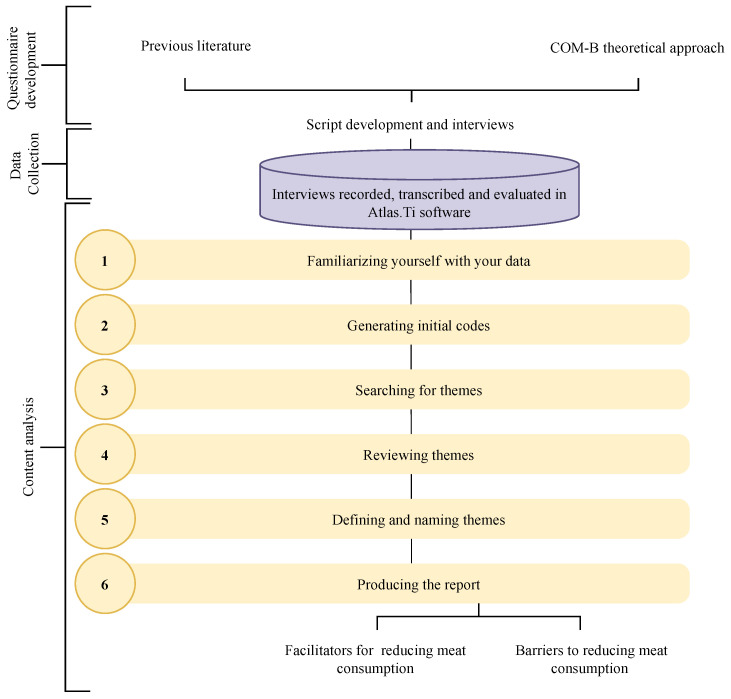
Content analysis in six steps for pattern identification, based on studies (adapted from Braun & Clarke [32].

**Figure 4 foods-12-00188-f004:**
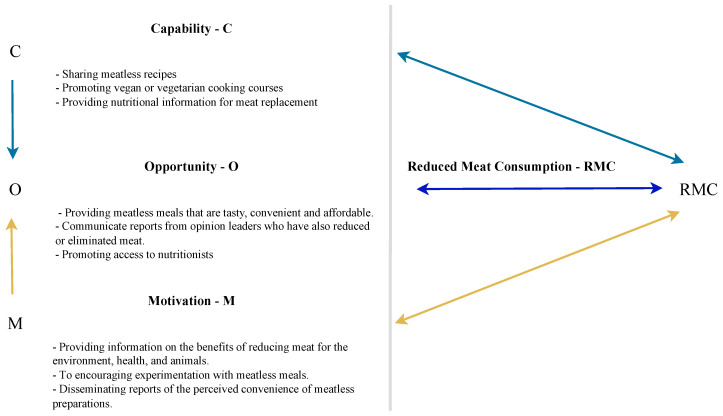
Proposed actions by external agents.

**Figure 5 foods-12-00188-f005:**
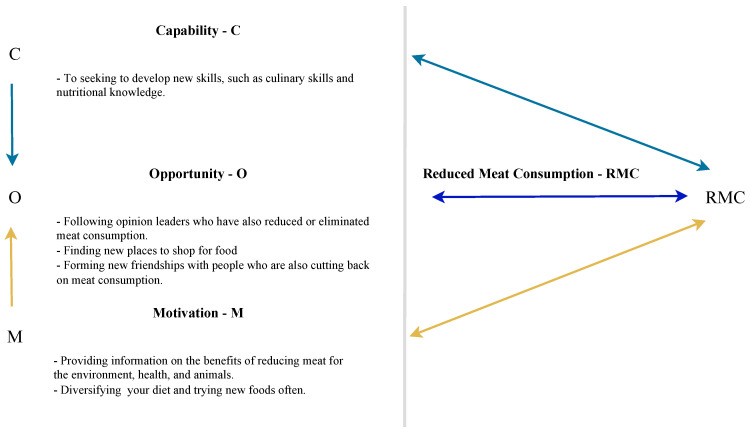
Proposed actions for consumers.

## Data Availability

All data used in this research are available in the article and in the Appendix A. Any additional information may be requested from the corresponding author.

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
