# Peer review of "Consumer Behavior Concerning Meat Consumption: Evidence from Brazil"

_foods, 2023, doi:10.3390/foods12010188_

Round 1

Reviewer 1 Report

The paper reports on a qualitative study examining facilitators of and barriers to the reduction of consumption in Brazil. Semi-structured interviews were conducted with 18 consumers who reduced their meat consumption. Content analysis was carried out to identify facilitators and barriers.

The topic is highly relevant for various disciplines and has numerous policy implications. However, the paper has several shortcomings which need to be addressed:

1.

The authors mention several times that Brazil is the fifth-largest meat-consuming market in the world. First, they should include a reference. Second, they should state whether the meat consumption ranking is per capita. If not, they should state the per capita statistics.

2.

“As a result, there are specific journals (for example, Foods, Appetite, Meat Sci- 41 ence, Food Quality and Preference, Trends in Food Science & Technology and the British  42 Food Journal) and special issues dedicated to eating behavior related to meat substitute  43 products. For example, Food Science and Technology - Innovative Trends in Plant-Based  44 Meat Alternatives (2022). 45”

I suggest the authors delete this section as it does not fit well into a scientific paper.

3.

“As far as is known, such studies have  67 not been conducted in Brazil, a developing country, where meat consumption is signifi- 68 cant [23].”

Unclear what significant refers to. Significant tends to indicate statistical significance. I suggest replacing it with other, more appropriate word.

I suggest replacing “hot topic” with a synonymy used in scientific writing.

4.

“The change in consumer behavior identified in the literature shows that the cultural  72 issues of meat consumption may be less evident  [16].”

It is unclear what “cultural issues” refer to in this sentence.

5.

“In this respect, the present study aims to apply and analyze, from the theoretical  69 perspective of the COM-B method [24],”

The main problem of the paper is in the use of COM-B method, or “theoretical perspective”, as the authors put it. COM-B model is first mentioned in the Abstract, but it is suggested that acronyms are not used the first time without explaining them. Authors then continue using the acronym for several times in the main text before the reader can read the author’s explanation.

More importantly, it is unclear to the reader how a COM-B method represent a theoretical perspective. As Figure 1 indicates, it is merely a Capability, Opportunity, Motivation schemata. This alone hardly constitutes a theoretical perspective, in my opinion. However, if it does, the authors should explain in the paper how they see it as a theoretical perspective. COM-B can namely be used in explaining various types of food and non-food related behaviours, and its significance is not really mentioned in the text, outside the fact that it refers to three “usual suspects” (C-O-M).

6.

“. The first version of the questionnaire was sub- 10 mitted for a pre-test to assess the reliability and validity of the research instruments before  108 the final application.”

How were reliability and validity assessed?

7.

“Intentional sampling was employed in this work, using the researchers’ contact net- 126 work, dissemination  on social media, and sending invitations to universities located in  127 Brazil.”

Is this intentional or snowball sampling?

8.

“This section may be divided by subheadings. It should provide a concise and precise  173 description of the experimental results, their interpretation, as well as the experimental  174 conclusions that can be drawn. 175”

??

9.

“On the other hand, other studies have already suggested that con- 478 sumers are increasingly aware of the relationship between health and food [38] and that  479 evidence has shown that the promotion of health benefits can be one of the strongest ar- 480 guments for the promotion of low-meat diets [20]. Given this finding, it is suggested that  481 future  efforts  to  reduce  meat  should  include  disseminating  clear  and  consistent  infor- 482 mation on the various impacts related to food choices.”

“Given this finding” and similar phrases should relate to author’s study findings and make recommendations from their findings, not (so much from) others.

10.

“people who were retired or over 60 years of age had less of an appetite for meat  498 (Kemper,  2020).”

Reference style.

11.

“As a theoretical contribution, this work reinforced previous research findings by con- 530 firming the presence of a set of facilitators and barriers for the reduction of meat consump- 531 tion  previously  suggested  in  the  literature. This  research  also  proposed  and  discussed  532 eight new potential facilitators and four new potential barriers to reducing meat consump- 533 tion, which is one of the main contributions of this article.”

I suggest repeating the eight and four. However, does this really constitute originality, i.e. new findings? Another three interviewees might mention another one, and so on. Novel findings of the study should be more explicit and meaningful, then mentioned and then discussed, and they should be novel also in the sense of building on existing theory.

12.

“This study has some limitations, mainly due to its sample size, which requires care  548 when making generalizations.”

Can 18 interviewees represent any generalization at all?

13.

“Given the importance of Brazil, considered the fifth largest meat consumer market  553 in the world,”

I suggest deleting the first part of the sentence.

14.

“Federal University of Parana –  Universal project  570 No. 404916/2018-0-Univ.    571”.

This needs to be put into a sentence form.

15.

A few references on the frequency of meat consumption and determinants of meat intake in the Theoretical part and/or Discussion would put the results into a broader context. A few recent examples include:

Koch, F.; Heuer, T.; Krems, C.; Claupein, E. Meat consumers and non-meat consumers in Germany: A characterization based on rresults of the German National Nutrition Survey II.J. Nutr. Sci.2019,8, e21

Einhorn, L. Meat consumption, classed? Osterr. Z. für Soziologie 2021, 46, 125–146.

Kirbiš, A.; Lamot, M.; Javornik, M. The Role of Education in Sustainable Dietary Patterns in Slovenia. Sustainability 2021, 13, 13036. https://doi.org/10.3390/su132313036

16.

The paper needs to be thoroughly language-edited. For example:

Abstract:

“To contribute to the 24 advancement of knowledge of meat-reduced diets,”

Change “of” meat to “on” meat.

17.

I suggest the authors thoroughly update the paper, as the topic of the paper is of the most importance to humans, animals and the planet.

Author Response

Response to Editor and Reviewers

Editor-in-chief

Foods (MDPI)

--

Thank you so much for reviewing our manuscript entitled “Consumer behavior concerning the meat consumption: evidence from Brazil” (ID foods-2050557) and inviting us to review and resubmit our manuscript to the Foods. Based on the reviewer’s suggestions, we thoroughly revised our manuscript and believe the manuscript is significantly improved. We are grateful to you and the reviewers for the positive feedback and constructive comments. We have organized this response letter around all the comments we received. Following the letter narrative, we provided responses outlining the detailed changes made to the manuscript. We believe the revised paper is a considerable improvement after addressing all the feedback from the reviewers. We have made every effort to response all the comments we received, and we hope the manuscript is now suitable for publication in the Foods. Please feel free to contact us with any questions.

We greatly appreciate your kind consideration of our manuscript, and look forward to your decision.

Thank you so much!

sincerely,

Claudimar Pereira da Veiga on behalf the authors.

----

#Reviewer 1

The paper reports on a qualitative study examining facilitators of and barriers to the reduction of consumption in Brazil. Semi-structured interviews were conducted with 18 consumers who reduced their meat consumption. Content analysis was carried out to identify facilitators and barriers.

The topic is highly relevant for various disciplines and has numerous policy implications. However, the paper has several shortcomings which need to be addressed:

Response: Thank you very much for your constructive feedback and suggestions. Your comments and advice shed light on our research. We believe that by addressing their recommendations, point by point, the article has become better. We are grateful to you for that.

1. The authors mention several times that Brazil is the fifth-largest meat-consuming market in the world. First, they should include a reference. Second, they should state whether the meat consumption ranking is per capita. If not, they should state the per capita statistics.

Response: Many thanks for this observation. Brazil is 3rd in the world ranking for meat consumption. Brazil is a developing country where meat consumption is significant (24.443 kg per capita), second only to Argentina (36.035 kg per capita) and the United States (25.969 kg per capita): available at: https://www.em .com.br/app/noticia/economia/2022/02/24/internas_economia,1347560/brasil-e-terceiro-consumidor-de-carne-bovina-no-mundo.shtml.

OECD (2022), Meat consumption (indicator). DOI: 10.1787/fa290fd0-en (Accessed on 02 December 2022). Available in: https://data.oecd.org/agroutput/meat-consumption.htm

Adjustments were made to the text, emphasizing that Brazil is a large meat consumer.

Country

Kilograms/capita

Argentina

36.035

United States

25.969

Brazil 

24.434

Israel  

24.105

Chile

21.831

SOURCE: OECD (2022), Meat consumption (indicator). doi: 10.1787/fa290fd0-en (Accessed on 02 december 2022). https://data.oecd.org/agroutput/meat-consumption.htm

2. “As a result, there are specific journals (for example, Foods, Appetite, Meat Science, Food Quality and Preference, Trends in Food Science & Technology and the British  Food Journal) and special issues dedicated to eating behavior related to meat substitute products. For example, Food Science and Technology - Innovative Trends in Plant-Based Meat Alternatives (2022). ”

I suggest the authors delete this section as it does not fit well into a scientific paper.

Response: Thanks for your suggestion. The text has been deleted from the manuscript.

3. “As far as is known, such studies have 67 not been conducted in Brazil, a developing country, where meat consumption is significant [23].”

Unclear what significant refers to. Significant tends to indicate statistical significance. I suggest replacing it with other, more appropriate word.

Response: Thank you very much for this important observation. The text has been changed, and we believe it got better. Adjustments are highlighted in red in the text. The details are the same as those answered in suggestion number 1.

I suggest replacing “hot topic” with a synonymy used in scientific writing.

Response: Thanks for this suggestion. The adjustments were made, and the text was as follows: “In light of these issues, this has been one of the themes of recent research in food science.”

4. “The change in consumer behavior identified in the literature shows that the cultural issues of meat consumption may be less evident [16].”

It is unclear what “cultural issues” refer to in this sentence.

Response: Thanks for your observation. New sentence: “The change in consumer behavior identified in the literature shows that the cultural domain of meat consumption may be less evident”.

5. “In this respect, the present study aims to apply and analyze, from the theoretical 69 perspective of the COM-B method [24],”

The main problem of the paper is in the use of COM-B method, or “theoretical perspective”, as the authors put it. COM-B model is first mentioned in the Abstract, but it is suggested that acronyms are not used the first time without explaining them. Authors then continue using the acronym for several times in the main text before the reader can read the author’s explanation.

Response: Thanks for this observation. The text has been changed and the acronyms explained the first time it appears (abstract). We believe that now it will be clearer for the reader.

More importantly, it is unclear to the reader how a COM-B method represent a theoretical perspective. As Figure 1 indicates, it is merely a Capability, Opportunity, Motivation schemata. This alone hardly constitutes a theoretical perspective, in my opinion. However, if it does, the authors should explain in the paper how they see it as a theoretical perspective. COM-B can namely be used in explaining various types of food and non-food related behaviours, and its significance is not really mentioned in the text, outside the fact that it refers to three “usual suspects” (C-O-M).

Response: Many thanks for these important comments. We hadn't thought about it from this angle. Indeed, the COM-B can namely be used in explaining various types of food and non-food related behaviors. So, we took advantage of the adjustments and clarified in the text the significance of the COM - B System. Therefore, the theoretical perspective was excluded from the manuscript. With this, we believe that the contribution became more evident.

 6. “. The first version of the questionnaire was submitted for a pre-test to assess the reliability and validity of the research instruments before the final application.”

How were reliability and validity assessed?

Response: Thank you for this observation. We apologize for this mistake. The pre-test was performed for content validation. This validation was necessary because the questionnaire was translated from another language. Adjustments have been made to the main text.

7. “Intentional sampling was employed in this work, using the researchers’ contact net- 126 work, dissemination on social media, and sending invitations to universities located in 127 Brazil.”

Is this intentional or snowball sampling?

Response: Thank you very much for this observation. We apologize for this error. The correct one is snowball sampling. The correction was made to the text

8. “This section may be divided by subheadings. It should provide a concise and precise description of the experimental results, their interpretation, as well as the experimental conclusions that can be drawn”

??

Response: Thank you very much for this observation; this sentence was confusing. Therefore, we prefer to exclude it from the text.

9. “On the other hand, other studies have already suggested that consumers are increasingly aware of the relationship between health and food [38] and that evidence has shown that the promotion of health benefits can be one of the strongest arguments for the promotion of low-meat diets [20]. Given this finding, it is suggested that future efforts to  reduce  meat  should  include  disseminating  clear  and  consistent  information on the various impacts related to food choices.”

“Given this finding” and similar phrases should relate to author’s study findings and make recommendations from their findings, not (so much from) others.

Response: Thank you very much for this observation. The finding shown is part of the study by Tucker, C. A. (2014). The correction was made in the text, and credit was given to the referred author. We apologize for our mistake.

10. “people who were retired or over 60 years of age had less of an appetite for meat (Kemper, 2020).”

Reference style.

Response: Thank you for showing us this mistake. The reference style has been corrected in the text.

11. “As a theoretical contribution, this work reinforced previous research findings by confirming the presence of a set of facilitators and barriers for the reduction of meat consumption previously suggested in the literature. This research also proposed  and  discussed  eight new potential facilitators and four new potential barriers to reducing meat consumption, which is one of the main contributions of this article.”

I suggest repeating the eight and four. However, does this really constitute originality, i.e. new findings? Another three interviewees might mention another one, and so on. Novel findings of the study should be more explicit and meaningful, then mentioned and then discussed, and they should be novel also in the sense of building on existing theory.

Response: Thank you very much for this comment. The eight new potential enablers and the four new potential barriers were specified again. We agree that the methodology limits the strength of the evidence of the new findings, so we highlight the study's limitations. We believe that by addressing this, our article has become better. Changes are highlighted in the text.

12. “This study has some limitations, mainly due to its sample size, which requires care when making generalizations.”

Can 18 interviewees represent any generalization at all?

Response: Thank you very much for notifying this. We made it even more evident in the study's limitations that generalizations should not be made because of the sample size.

13. “Given the importance of Brazil, considered the fifth largest meat consumer market in the world,”

I suggest deleting the first part of the sentence.

Response: The first part of the sentence was deleted. Thank you for this suggestion.

14. “Federal University of Parana – Universal project 570 No. 404916/2018-0-Univ”.

This needs to be put into a sentence form.

Response: Thanks for this observation. The adjustment was made in the text.

15. A few references on the frequency of meat consumption and determinants of meat intake in the Theoretical part and/or Discussion would put the results into a broader context. A few recent examples include:

Koch, F.; Heuer, T.; Krems, C.; Claupein, E. Meat consumers and non-meat consumers in Germany: A characterization based on results of the German National Nutrition Survey II.J. Nutr. Sci.2019,8, e21

Einhorn, L. Meat consumption, classed? Osterr. Z. für Soziologie 2021, 46, 125–146.

Kirbiš, A.; Lamot, M.; Javornik, M. The Role of Education in Sustainable Dietary Patterns in Slovenia. Sustainability 2021, 13, 13036. https://doi.org/10.3390/su132313036

Response: Thank you very much for shedding light on our research. The studies by Koch et al. (2021) and Einhorn were included in this paper. We are on a new project, and all these references will be helpful to bring contributions to the new study. We are grateful to you for that.

16. The paper needs to be thoroughly language-edited. For example:

Abstract:

“To contribute to the advancement of knowledge of meat-reduced diets,”

Change “of” meat to “on” meat.

Response: Thanks for this comment. The article was revised by an English writing professional. We believe the issues have been resolved.

17. I suggest the authors thoroughly update the paper, as the topic of the paper is of the most importance to humans, animals and the planet.

Response: Thanks for this suggestion. Based on this comment, the importance of the topic for humans, animals and the planet were highlighted in the introduction.

“Given the impact of meat production and consumption on environmental sustainability, animal suffering, and human health, the recent attention of researchers on the topic is not surprising. Regarding the eating behavior of meat consumption, recent research has pointed out some problems associated with environmental degradation, health problems, and compromised animal welfare”.

We also highlight this topic for suggestions for future work.

*Dear reviewer

We are very grateful for your attention in reading our work and for all suggested improvement points. Although we have submitted several academic papers for renowned journals over time, we can say with certainty that it was the best review we have ever received. We believe that by addressing your comments and suggestions, our revised manuscript is significantly improved. Receive our sincere thanks.

Reviewer 2 Report

Dear authors,

I find this study comprehensive and well-structured. The topic it deals with is a current and relevant issue that definitely needs to be explored in every aspect. 

Introduction: The introduction is well structured and the supporting literature seems relevant and current. My advice is only to further enhance your research question in order to make it more "intriguing" to the reader.

Materials and methods: this section is also well structured. However, I would ask you for some additional information on the model used. Why did you choose this one out of all?

Results and discussions: These two sections are satisfactory. They explain all the factors in detail and the facts are clearly stated.

Conclusions: implications are highlighted, however, I think they can be further highlighted, as well as implications for future studies.

Overall, it is a good study. I wish you well in future research.

Author Response

Dear Reviewer

Thank you so much for reviewing our manuscript entitled “Consumer behavior concerning the meat consumption: evidence from Brazil” (ID foods-2050557) and inviting us to review and resubmit our manuscript to the Foods. Based on the reviewer’s suggestions, we thoroughly revised our manuscript and believe the manuscript is significantly improved. We are grateful to you and the reviewers for the positive feedback and constructive comments. We have organized this response letter around all the comments we received. Following the letter narrative, we provided responses outlining the detailed changes made to the manuscript. We believe the revised paper is a considerable improvement after addressing all the feedback from the reviewers. We have made every effort to response all the comments we received, and we hope the manuscript is now suitable for publication in the Foods. Please feel free to contact us with any questions.

We greatly appreciate your kind consideration of our manuscript, and look forward to your decision.

Thank you so much!

sincerely,

Claudimar Pereira da Veiga on behalf of the authors.

--

Dear authors,

I find this study comprehensive and well-structured. The topic it deals with is a current and relevant issue that definitely needs to be explored in every aspect. 

Response: Thank you very much for your kind words. We really appreciate it.

Introduction: The introduction is well structured and the supporting literature seems relevant and current. My advice is only to further enhance your research question in order to make it more "intriguing" to the reader.

Materials and methods: this section is also well structured. However, I would ask you for some additional information on the model used. Why did you choose this one out of all?

Results and discussions: These two sections are satisfactory. They explain all the factors in detail and the facts are clearly stated.

Conclusions: implications are highlighted, however, I think they can be further highlighted, as well as implications for future studies.

Overall, it is a good study. I wish you well in future research.

Response: Thanks a lot for your comments. This motivates us more and more to face the challenges and difficulties in carrying out research, especially in Brazil, where we do not have the necessary investment. We are grateful to you for everything.

Round 2

Reviewer 1 Report

The paper is much improved now.

"Although we have submitted several academic papers for renowned journals over time, we can say with certainty that it was the best review we have ever received."

The reviewer appreciates the positive feedback on the usefulness of the review.

Lastly, I would still suggest that the authors include all three references that were suggested on the frequency of meat consumption and determinants of meat intake, whether in the Theoretical part or the Discussion section. The authors decided to leave out a reference from a MDPI journal, although including it would indicate to the reader interested in the topic that the topic of determinants of meat consumption has been published in and can be read more about in MDPI journals (the majority of present references in the paper are from Appetite journal etc.).

Author Response

Dear Editor and Reviewer

Thank you so much for reviewing our manuscript entitled “Consumer behavior concerning the meat consumption: evidence from Brazil” (ID foods-2050557) and inviting us to review and resubmit our manuscript to the Foods. Based on the reviewer’s suggestions, we thoroughly revised our manuscript and believe the manuscript is significantly improved. We are grateful to you and the reviewers for the positive feedback and constructive comments. We have organized this response letter around all the comments we received. Following the letter narrative, we provided responses outlining the detailed changes made to the manuscript. We believe the revised paper is a considerable improvement after addressing all the feedback from the reviewers. We have made every effort to response all the comments we received, and we hope the manuscript is now suitable for publication in the Foods. Please feel free to contact us with any questions.

We greatly appreciate your kind consideration of our manuscript, and look forward to your decision.

Thank you so much!

sincerely,

Claudimar Pereira da Veiga on behalf of the authors.

# Reviewer

# The paper is much improved now.

Response: Thank you very much for your words. We really appreciate it. We are grateful to you for bringing light to our work.

# "Although we have submitted several academic papers for renowned journals over time, we can say with certainty that it was the best review we have ever received."

# The reviewer appreciates the positive feedback on the usefulness of the review.

Response: This only reinforces the reviewers' work's importance for the research's evolution. At the article's end, we give the reviewers special thanks.

# Lastly, I would still suggest that the authors include all three references that were suggested on the frequency of meat consumption and determinants of meat intake, whether in the Theoretical part or the Discussion section.

Response: We insert the three references in the text. We believe the article level has considerably improved when addressing frequency and determinants of meat intake related to reduced meat consumption and sustainability issues. The inserted part is highlighted in red color at the end of the introduction.

# The authors decided to leave out a reference from a MDPI journal, although including it would indicate to the reader interested in the topic that the topic of determinants of meat consumption has been published in and can be read more about in MDPI journals (the majority of present references in the paper are from Appetite journal etc.).

Response: We have inserted new articles from Base MDPI, such as Foods, and Sustainability into the manuscript.

The writing of the article was again revised. We believe the writing issues have been resolved.

Again, many thanks for your advice and guidance.

Claudimar Pereira da Veiga on behalf of the authors.